# A New Compartment Model of COVID-19 Transmission: The Broken-Link Model

**DOI:** 10.3390/ijerph19116864

**Published:** 2022-06-03

**Authors:** Yoichi Ikeda, Kenji Sasaki, Takashi Nakano

**Affiliations:** 1Department of Physics, Faculty of Science, Kyushu University, Fukuoka 819-0395, Japan; 2Center for Infectious Disease Education and Research, Osaka University, Osaka 565-0871, Japan; kenjis@cider.osaka-u.ac.jp (K.S.); nakano@rcnp.osaka-u.ac.jp (T.N.); 3Research Center for Nuclear Physics, Osaka University, Osaka 567-0047, Japan

**Keywords:** COVID-19, epidemic model, compartment model, delta variant, omicron variant

## Abstract

We propose a new compartment model of COVID-19 spread, the broken-link model, which includes the effect from unconnected infectious links of the transmission. The traditional SIR-type epidemic models are widely used to analyze the spread status, and the models show the exponential growth of the number of infected people. However, even in the early stage of the spread, it is proven by the actual data that the exponential growth did not occur all over the world. We presume this is caused by the suppression of secondary and higher-order transmissions of COVID-19. We find that the proposed broken-link model quantitatively describes the mechanism of this suppression, which leads to the shape of epicurves of confirmed cases are governed by the probability of unconnected infectious links, and the magnitudes of the cases are proportional to $\exp\left(R_{0}\right)$ in each infectious surge generated by a virus of the basic reproduction number $R_{0}$, and is consistent with the actual data.

## 1. Introduction

Since the first case of novel coronavirus infectious disease (COVID-19) was reported in Wuhan, China, COVID-19 has spread all over the world. In order to save the lives from the threat of COVID-19 and maintain social activities from the viewpoint of economy, it is vital to ascertain the status of the spread as accurate as possible.

The SIR (susceptible-infected-removed) model and its family members such as the SEIR (susceptible-exposed-infected-removed) model have been widely used compartment models for trying to describe the projection of COVID-19 spread. The SIR model was first applied to the plague in the island of Bombay over the period December 1905 to July 1906 [1]. The first-order coupling between susceptible and infected people was assumed, and this treatment was justified for the plague mediated by carrier rats which form a mean field of the plague, and thus the susceptible people have an equal probability of being infected. Indeed, the calculated epidemic curve during the period of epidemic roughly corresponded to the reported numbers. One of the typical features of the SIR-type models is that the models predict the exponential growth of the number of infected and removed people (confirmed cases) for the early stage of the spread [2].

There have been numerous studies to forecast COVID-19 spread in various regions using the SIR-type models since the COVID-19 outbreak in Wuhan, China, in December 2019 [3,4,5,6,7,8,9]. The model projections overestimated the number of confirmed cases due to the exponential growth, and the discrepancy between the model predictions and actual data could be attributed to the restriction of social activities such as the lockdowns of cities. Studies to evaluate the effects of policies were conducted [10,11,12,13,14], but the results have depended on the models adopted and the period to be analyzed [15,16,17,18]. This suggests that one should take into account the transmission mechanism of coronavirus spread not through a mean-field corresponding to the exponential growth of infected people but through a contact process.

Meanwhile, the indicator of the spread rate, what is called the K-value, defined by $K\left(t\right)=1-R\left(t-7\right)/R\left(t\right)$ with $R\left(t\right)$ being the cumulative number of confirmed cases at day t from a reference date as shown in Table 1, exhibits nonexponential growth of $R\left(t\right)$ even in the early stage of the spread but exhibits approximate linear decrease of the K-value transition universally in many countries [19]. The linearly decreasing behavior of the K-value transition was well reproduced by the phenomenologically developed constant attenuation model [19], whereas $R\left(t\right)$ is expressed as $R\left(t\right)=R\left(0\right)\exp\left(a\left(t\right)\;t\right)$, and $a\left(t\right)$ is defined by the geometric progression $a\left(t\right)=\exp\left[-\left(1-k\right)\right]a\left(t-1\right)$ with a constant attenuation factor k. Based on the constant attenuation model, it was found that $R\left(t\right)$ follows the Gompertz curve [20,21,22].

In this paper, we propose a new compartment model, the *broken-link model*, in order to comprehensively understand why the COVID-19 transmission follows the Gompertz curve. The model is naturally derived from the observation of suppression of COVID-19 transmission in the secondary cases generated by the primary ones [23]. We also applied the model to the epidemic surges generated by the delta (δ) and omicron (ο) variants in Japan, South Africa, the United States, France and Denmark. These countries are chosen by focusing on the variants that caused each surge. For South Africa and the U.S., they are examples to see the transmission of the ο variant from equilibrium states with small and large numbers of daily confirmed cases, respectively. The others are to see the effects of the change in variants from δ to ο, and more details will be seen in the Results and Discussion sections.

## 2. Materials and Methods

Recalling that the coronaviruses spread through a contact process and the suppression of secondary and higher-order transmissions, we now derive the broken-link model. For this purpose, we start with the SIR model. In the SIR model, we partition the total population into three compartments, susceptible, infected and removed individuals, and represent the numbers of three compartments at time t by $S\left(t\right)$, $I\left(t\right)$ and $R\left(t\right)$. The SIR model is then described as coupled ordinary differential equations (ODEs),
(1)$$\frac{dS}{dt}=-\beta SI,\;\frac{dI}{dt}=\gamma \left(R_{0}\frac{S}{N}-1\right)I,\;\frac{dR}{dt}=\gamma I,$$
where β and γ are contact and removal rates of infections, respectively. The basic reproduction number is denoted by $R_{0}=\beta N/\gamma$ with N being the total population number. When the cumulative number of infected persons $R\left(t\right)$ is much less than the total population, $S\left(t\right)$ can be approximated by N. Then one finds the exponential growth of $I\left(t\right)$ and $R\left(t\right)$ which cannot be inevitable unless the contact and removal rates are assumed to be constant in the period of epidemic.

One of the good indicators to find out the behavior of COVID-19 transmission is the K-value. The analysis using the K-value has revealed that the cumulative number of confirmed cases $R\left(t\right)$ follows the Gompertz curve even in the early stage of the spread, when herd immunity has not been achieved at all. A natural reason is that COVID-19 spread through contact and/or local droplet processes. As reported in Nishiura et al. [23] in their study of COVID-19 secondary transmission, the secondary transmission generated from the primary cases in non-close environments is highly suppressed.

We model the suppression of the secondary and higher-order transmission in terms of compartment models. According to Nishiura et al. [23], all the transmission links are not connected to the next generation. When the links are connected through the probability k, in other words through the broken-link probability $\left(1-k\right)$, the subsequent transmissions are not generated. Therefore, we cut these contributions from a transmission tree as shown in Figure 1a.

Now, we formulate the *broken-link model*. In addition to the S, I and R compartments, it is natural to introduce the S′ (temporary removed) compartment due to unconnected transmission links as shown in Figure 1b. The time evolutions of $S\left(t\right)$, $I\left(t\right)$, and $R\left(t\right)$ are respectively expressed by the following coupled ODEs:(2)$$S\left(t\right)=S_{0}k^{t},$$
(3)$$\frac{dI\left(t\right)}{dt}=\gamma \left(R_{0}\frac{S\left(t\right)}{N}-1\right)I\left(t\right),$$
(4)$$\frac{dR\left(t\right)}{dt}=\gamma I\left(t\right),$$
where $S_{0}$ in Equation (2) is the number of susceptible people who are potentially under the threat of transmissions in each epidemic wave, and the number of temporary removed people is given by $S^{\prime }\left(t\right)=S_{0}\left(1-k^{t}\right)$. It is worth mentioning that we neglect the tiny contribution to decrease in S through the bilinear term −βSI in Equation (1). Although the bilinear term gives less than 1% contribution, since the number of infected people is two or three orders of magnitude smaller than the total population, this is the limitation of this model when I becomes comparable with S.

The analytic solutions of $I\left(t\right)$ and $R\left(t\right)$ are found as
(5)$$I\left(t\right)=R_{0}N_{\infty }k^{t}e^{-R_{0}k^{t}}$$
(6)$$R\left(t\right)=N_{\infty }e^{-R_{0}k^{t}}$$
with γ=−lnk in Equations (3) and (4), and $N_{\infty }=R\left(0\right)\exp\left(R_{0}\right),$ which represents the cumulative number of infected people in each infection wave generated by a coronavirus with the basic reproduction number $R_{0}$ (see also Table 2). In Equation (6), we can see that the cumulative number $R\left(t\right)$ satisfies the Gompertz curve. It also turns out that the probability k is equivalent to the constant attenuation factor [19], so that the phenomenologically introduced constant attenuation is consistent with the suppression of transmissions due to the unconnected transmission links.

There are two remarkable findings based on the broken-link model. First, the shapes of epicurves of the daily and cumulative confirmed cases are governed by the value of the probability k, and the magnitudes of the cases are proportional to $\exp\left(R_{0}\right)$ in each infection wave. Second, the basic reproduction number $R_{0}$ is inversely proportional to $-\ln k≅\left(1-k\right)$, that is, to the broken-link probability when k is close to one. Therefore, even though only the small regional difference in the probability k is obtained from the actual data, the orders of magnitudes of the confirmed cases can be largely different.

## 3. Results

The surges of COVID-19 occurred in various countries. We investigate the structure of each surge of COVID-19 assuming the broken-link model. All the data are taken from the COVID-19 Data Repository of the Center for Systems Science and Engineering (CSSE) at Johns Hopkins University [24]. In the analysis, the peak structure appears in daily confirmed cases as a counterpart of the Gompertz function in the cumulative number. This peak is called a *wave* in this article. The constant trend in daily confirmed cases is described as a *baseline* that corresponds to the endemic spread and yields the linear trend in the cumulative number.

### 3.1. The δ Epidemic Surge in Japan

We first look at the surge caused by the δ variant in Japan from late June to the end of September 2021. The epicurve in Figure 2a is decomposed into three partial waves and a baseline component. This decomposition is validated from the behavior of the K-value transition in Figure 2b because there are three peaks after quick reduction as 1/t, which indicates the existence of a baseline in daily confirmed cases. Based on this fact, the cumulative number is fitted by three Gompertz curves and a baseline. The results reproduce both the number of daily confirmed cases and the K-value very well. The fit parameters for each Gompertz curve in the δ surge are summarized in Table 3.

### 3.2. The ο Epidemic Surge in Japan

The trend in confirmed cases and the K-value for the ο surge in Japan are shown in Figure 3. The first infected person with ο variant as a community-acquired infection was reported on 22 December 2021 in Osaka prefecture and then ο variant spiked nationwide. As seen in Figure 3, the decomposition of the ο surge into two partial waves was justified by the trend in the K-value.

The fit parameters of the Gompertz curves for the ο surge in Japan are summarized in Table 4. The broken-link probability $\left(1-k\right)$ is smaller than that in the δ surge, which implies that the subsidence of the ο surge is slower than that for the δ surge. The cumulative number of confirmed ο cases in Japan is predicted to be about 5 times larger than that of the δ case.

### 3.3. The Status of the δ and ο Surges in Other Countries

In this subsection, we survey the status of the δ and ο surges in the other countries. In South Africa, where the ο variant was reported in the world for the first time, the surge emerged on 23 November 2021 as shown in Figure 4a. The wave in South Africa passed a peak at mid-December 2021. In U.S., the ο surge started rising at beginning of December 2021 with huge infectivity as shown in Figure 4b. In France, shown in Figure 4c, 2 partial waves were confirmed in the epicurve and the magnitude of the 2nd wave was much larger than that of 1st one. In the case of Denmark, where a different kind of ο variant was reported to spread, we were able to confirm the existence of three partial waves in the surge from November 2021 to mid-February 2022 in Figure 4d. Again, we easily see that the number of daily cases at peak was increasing in each wave. The results of the fit parameters are summarized in Table 5.

## 4. Discussion

As a new compartment model of COVID-19, we have proposed the broken-link model, where the suppression of secondary and higher-order transmissions is taken into account. Contrary to the SIR-type models, the coupling between susceptible S and infected I becomes time dependent in the broken-link model. The model predicts the Gompertz curve for the cumulative number of confirmed cases, which is consistent with the observations shown in Figure 2, Figure 3 and Figure 4. In the model, the shape of epicurves corresponds to the probability k, and the magnitude is proportional to $\exp\left(R_{0}\right)$ in which the basic reproduction number $R_{0}$ is obtained as $R_{0}=-a/\ln k≅a/\left(1-k\right)$ for k≅1 with a constant a, which is determined by the fit. Therefore, the small regional difference of the probability k observed in Section 3 is enhanced in the numbers of daily and total confirmed cases. Shown in Figure 5 is the predicted k dependence of the number of the daily cases at peak in the model.

From Nakano et al. [19], the mean value of the probability k in Japan was found to be k=0.92 (8% for the broken-link probability), which is also consistent with the 1st partial waves in the δ surge as shown in Table 3. On the other hand, for example in France, the broken-link probability was approximately 30% smaller as shown in Table 5. This difference gives about 12 and 17 times more daily cases at a peak and more cumulative cases than those in Japan, respectively. The regional k difference would be attributed to the immune response to coronaviruses [25]. Indeed, due to double vaccination, about 20% and 30% increases in the broken-link probability were observed for the 2nd and 3rd δ partial waves in Japan, respectively.

It is notable that the onset of epidemic surges and even partial waves has synchronized the appearance of new variants of coronaviruses from country to country. In Japan, the genomic surveillance by NIID (the National Institute of Infectious Diseases) [26] showed that the δ surge in Figure 2 was caused mainly by AY.29 and following AY.29.1 in terms of PANGO (Phylogenetic Assignment of Named Global Outbreak) Lineages [27,28]. It is interpreted that the temporary removed susceptible people from the transmission links of the α variant, which was already spread nationwide before the δ variant emerged, were brought back under the threat of the δ surge. The emergence of a new partial wave was also attributed to the appearance of new variants having a stronger transmissibility than the others.

It is also important to investigate the situation in other countries with respect to the genomic surveillance. In South Africa, shown in Figure 4a, BA.1 cases were dominant during the ο surge and then BA.2 cases gradually increased from mid-December 2021. In the U.S., as shown in Figure 4b, the ο surge was caused by BA.1 and BA.1.1, for which transmissibility rates are expected to be similar. Thus, the fit with a single Gompertz curve worked very well. For the case of France, shown in Figure 4c, the 1st and 2nd waves were caused by the δ and ο variants, respectively. This fact was able to be confirmed by the results of $R_{0}$ given in Table 5 because the $R_{0}$s in the 1st and 2nd waves are consistent with the typical values for the δ [29] and ο [30] variants, respectively.

The situation is slightly complicated in Denmark shown in Figure 4d. The 1st wave was generated by the δ variant, and the others were caused by the ο variant. The genomic surveillance report from *outbreak.info* [31] indicated that BA.2 cases emerged from mid-December 2021 and became dominant at the end of January 2022. According to the report [31], we find that two points that the 2nd and 3rd waves were caused respectively by BA.1 and BA.2 and that BA.2 has enough strong infectivity to generate a new wave in daily confirmed cases.

As shown above, the broken-link model naturally derived from the observation of the suppression of higher-order transmissions works well for short-term forecasts. In the actual data, there exists a constant trend (baseline) after a surge converges, and the description of baselines is out of the model space. So, it is a challenging future problem to extend the model to describe a long-term forecast over a few months.

## 5. Conclusions

We proposed a new compartment model of COVID-19 spread, the broken-link model, which includes the effect from unconnected infectious links of the transmission. The model took into account the suppression of secondary and higher-order transmissions of COVID-19 in the traditionally used compartment model for COVID-19 spread analyses. The cumulative number of confirmed cases $R\left(t\right)$ in the model satisfies the Gompertz curve, whose parameters are characterized as the cumulative number of infected people $N_{\infty }$, the basic reproduction number $R_{0}$ and the connection probability of transmission links k, which was defined as the attenuation factor in a previous paper [19].

The model applied to the actual data for the epidemic surges of coronaviruses in Japan, South Africa, the United States, France and Denmark. From these results with the detailed genomic surveillance, we found that the onset of a partial wave has synchronized the appearance of new variants of coronaviruses and a scale of total infected people is closely related to the probability k. The typical value of k in Japan evaluated in this study is smaller than those in European countries for the δ surge, but it gets close to European ones for the o surge.

The proposed broken-link model is potentially applied not only to COVID-19 but to other epidemics since it is designed to describe the transmission through contact processes. One of our future targets is the transmission of influenza, and the results will be reported elsewhere.

## Figures and Tables

**Figure 1 ijerph-19-06864-f001:**
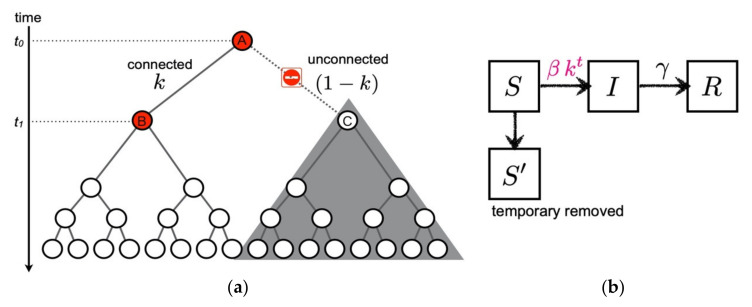
A suggested formula of the broken-link model. (**a**) Cartoon of the suppression of secondary infection in a transmission tree (k=1/2 case). When the transmission link from the primary infected individual A to the secondary candidate C is unconnected at time $t_{1}$, subsequent transmissions starting from C are not generated as denoted by the shaded area. (**b**) Compartments of the broken-link model. The temporary removed compartment S′ is introduced due to the suppression of the secondary and higher-order transmissions. Contrary to the SIR model, the coupling between susceptible S and infected I becomes time dependent in the broken-link model.

**Figure 2 ijerph-19-06864-f002:**
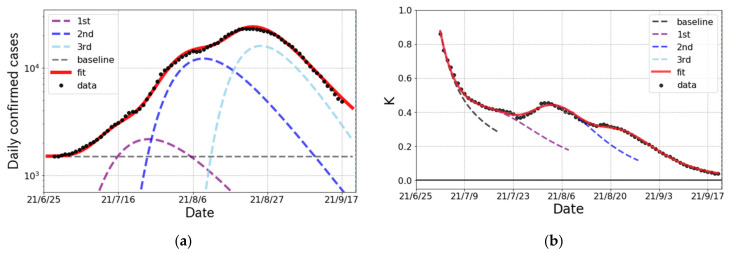
The epicurve of the δ surge of COVID-19 spread in Japan from June to September 2021. (**a**) The logarithmic plot of the number of daily confirmed cases (one week average) and the fit result (solid curve). The fit was performed with three partial waves and a baseline denoted by dashed lines. (**b**) The observed data and fit result of the K-value.

**Figure 3 ijerph-19-06864-f003:**
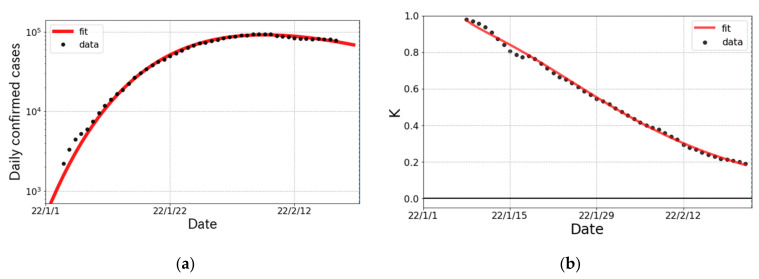
The ο surge in Japan from January to February 2022. (**a**) The logarithmic plot of daily confirmed cases (one week average) and fit result. The fit was performed with a single partial wave. (**b**) The observed data and fit result for the K-value.

**Figure 4 ijerph-19-06864-f004:**
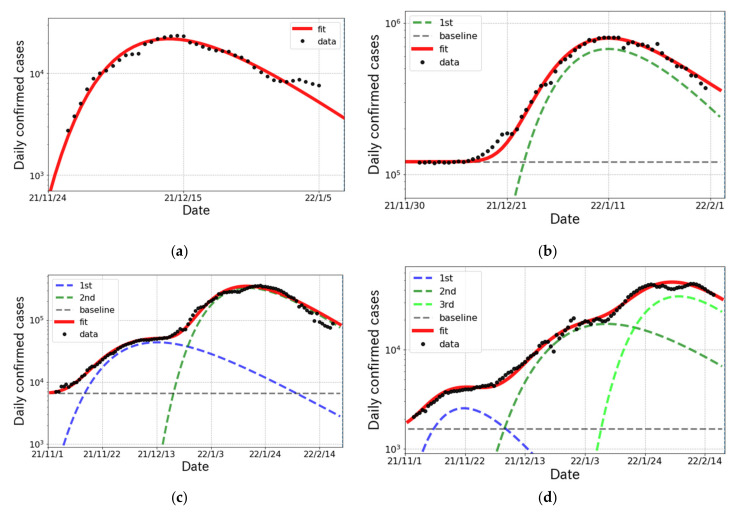
The logarithmic plot of daily confirmed cases (one week average) and the fit results for the δ and ο surges in South Africa, the United States of America, France and Denmark from November 2021 to February 2022. (**a**) South Africa; (**b**) United States; (**c**) France; (**d**) Denmark.

**Figure 5 ijerph-19-06864-f005:**
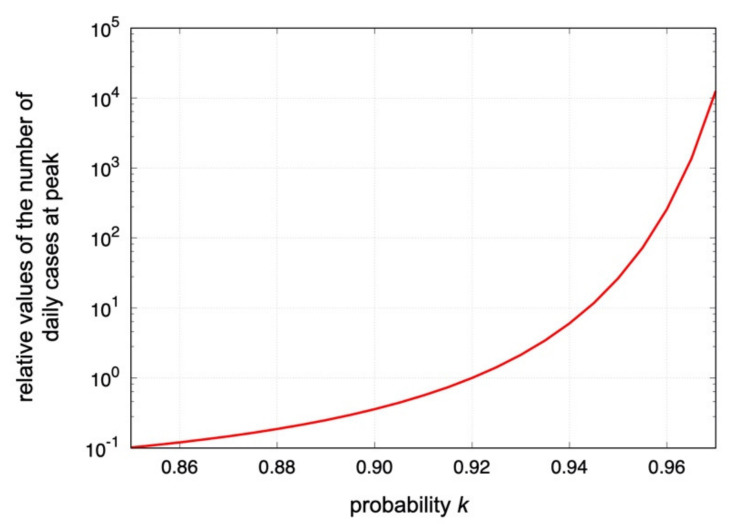
The predicted k dependence of the relative values of the number of daily confirmed cases at peak positions in the logarithmic scale. The relative value is normalized at k=0.92, which was obtained in the 1st epidemic surge in April 2020 in Japan [19]. The value is proportional to $-\left(\ln k\right)\exp\left(-a/\ln k\right)$ in the model. The case with a=0.5 is shown.

**Table 1 ijerph-19-06864-t001:** The parameters and functions for the K-value and phenomenological constant attenuation model.

Parameters/Functions	Descriptions
t	day
$R\left(t\right)$	cumulative number of confirmed cases
$K\left(t\right)=1-R\left(t-7\right)/R\left(t\right)$	K-value (indicator)
$a\left(t\right)=\exp\left[-\left(1-k\right)\right]a\left(t-1\right)$	geometric progression
k	constant attenuation factor

**Table 2 ijerph-19-06864-t002:** The parameters and functions for the broken-link model. For the basic reproduction number $R_{0}$, the constant parameter a is determined from the fit to the actual data. The analytic expressions of $R_{0}$, $N_{\infty }$ and γ are obtained by solving Equations (3) and (4).

Parameters/Functions	Descriptions
t	time
$S\left(t\right)$	number of susceptible people
$\;I\left(t\right)$	number of infected people
$R\left(t\right)$	cumulative number of confirmed cases
k	probability of connected transmission links
$R_{0}=-a/\ln k$	basic reproduction number
$N_{\infty }=R\left(0\right)\exp\left(R_{0}\right)$	cumulative number of infected people in each infection wave generated by $R_{0}$
γ=−lnk	removal rate from transmission trees

**Table 3 ijerph-19-06864-t003:** The parameters of the Gompertz curves in the δ surge in Japan. The $N_{\infty }$, $R_{0}$ and k are the cumulative number of infected people, basic reproduction number and connected probability of transmission links, respectively. The “shift” stands for the onset of a partial wave from the reference date (25 June 2021). The statistical errors evaluated by the jackknife method are represented in the parentheses.

Partial Wave	$N_{\infty }$	$R_{0}$	k	Shift (Days)
1st	75 (12) k	6.49 (20)	0.918 (4)	7.2 (3)
2nd	340 (23) k	6.98 (16)	0.907 (3)	24.5 (3)
3rd	375 (2) k	4.40 (15)	0.892 (1)	47.7 (3)

**Table 4 ijerph-19-06864-t004:** The parameters of the Gompertz curve for the ο surge in Japan. The definition of the parameters is the same as in Table 3, but the reference date is 1 January 2022.

Partial Wave	$N_{\infty }$	$R_{0}$	k	Shift (Days)
1st	4332 (6) k	10.4 (1)	0.944 (1)	−4.3 (1)

**Table 5 ijerph-19-06864-t005:** The parameters of the Gompertz curves for other countries from November 2021 to February 2022. The definition of the parameters is the same as in Table 3, but the reference dates are 24 November 2021, 30 November 2021, 1 November 2021 and 1 November 2021 for South Africa, United States, France and Denmark, respectively.

Region	Partial Wave	$N_{\infty }$	$R_{0}$	k	Shift (Days)
South Africa	1st	592 (1) k	8.98 (11)	0.905 (1)	−3.7 (2)
United States	1st	22,411 (269) k	9.68 (6)	0.922 (1)	13.6 (4)
France	1st	2233 (235) k	7.18 (22)	0.949 (2)	4.3 (6)
	2nd	13,330 (16) k	9.92 (1)	0.935 (1)	42.4 (1)
Denmark	1st	84 (1) k	6.16 (64)	0.924 (1)	−3.0 (6)
	2nd	1094 (124) k	9.93 (28)	0.956 (2)	19.4 (1.0)
	3rd	1401 (6) k	8.00 (25)	0.937 (1)	63.3 (5)

## Data Availability

Not applicable.

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
