# Peer review of "A New Compartment Model of COVID-19 Transmission: The Broken-Link Model"

_ijerph, 2022, doi:10.3390/ijerph19116864_

Round 1
Reviewer 1 Report
The manuscript has proposed a new compartment model of COVID-19 spread, the broken-link model, which includes the effect from unconnected infectious links of the transmission. As per the ms claims, the proposed model quantitatively describes the mechanism of suppression and is also consistent with the actual data.
The manuscript is prepared as well structured with smooth flow. I just have a minor observation that 14-references are too short to an article, so, recommend to strengthen the ms with further literature citations.
Author Response
Dear reviewer,
Thank you very much for reviewing our manuscript.
We appreciate the reviewer's substantial acceptance of our claims on our manuscript.
Please see the attachment.
Sincerely yours,
Yoichi Ikeda on the behalf of the authors

Reviewer 2 Report
This work proposed a new compartment model for COVID-19, named with the broken-link model. The authors' climes that the model quantitatively describes the mechanism of this suppression and is consistent with the actual data. Although this work has been conducted and tested. However, the numerical part has not been well explained. Therefore, I wrote some major and minor comments for better improvement, which should be addressed for possible acceptance in this journal.
Major:
- The manuscript requires significant proofreading and revision to improve the quality of English.
- The introduction is too short. The authors need to expand their writing with more references in this part.
- (lines: 35-44) In the introduction: Rewriting by separating the equations and shifted this part to the discussion section.
- Remove all equations from the text and make them separate with numbers.
- More explanation needs to show the analytic solutions for equations 3 and 4.
- The method section requires a more detailed explanation of how authors calculated the fitted.
- How did the authors choose (? and ?∞)?
- The results should include the effect of social distancing on the virus spread. A deeper investigation would be attractive during the lockdown period.
Minor:
- The abstract needs to be improved to include a brief methodology and some results.
Author Response
Dear reviewer,
We thank the reviewer very much for his/her detailed review and the careful reading of our manuscript.
Also, we really appreciate his/her valuable comments/suggestions.
We addressed all of the comments/suggestions and believe that the merits of the present work have become clearer through replying to the comments.
Please see the attachment.
We hope our manuscript is now ready for publication in Int. J. Environ. Res. Public Health.
Sincerely yours,
Yoichi Ikeda on the behalf of the authors

Reviewer 3 Report
The authors describe in this manuscript an interesting and increasingly intriguing topic to healthcare systems nowadays, the transmission models of COVID-19 Infection, throughout analyzing a novel hypothesis trying to explain the secondary and unconnected infection links of transmission of COVID-19, which have been until now relatively underestimated in the traditional analyses of COVID-19 Transmission models.
The title of the manuscript provides a well-defined introduction to the topic and includes a good choice of wording, which will be attractive to the reader (e.g. the phrase: broken link model)
The authors approached the topic in the abstract with a concise, Yet informative and minute description of the proposed new model of COVID-19 transmission.
The introduction of the manuscript starts with a very concise and summarized description of COVID-19 traditional models of spread analyses (SIR/SEIR). It is important to further explain the above-mentioned modalities comprehensively and in a well-detailed manner. The statistical aspects of COVID-19 transmission and the missed links in the traditional analyses were well presented. However, I recommend in this regard adding a list or table of definitions for the used indicators to help illustrate further details and the flow of reading (e.g. K-valu, R, t, etc…). Further in this part, the authors presented a very well-focused and relevant statement regarding the new proposed model of transmission. They applied this novel model of analysis to epidemic surges in Japan as well as other countries (South Africa, United States, France, Denmark). It would be recommended to add a statement regarding the criteria, for which these countries were chosen for the analysis.
Methodically, the authors presented their new proposed model of transmission in a very detailed and well-illustrated manner. An additional detailed statement regarding how the real-world transmission models follow the Gompertz curve in this part could be beneficial for the manuscript.
The results of the new proposed model were constructed and presented comprehensively with a suitable statistical evaluation.
The discussion, which follows reemphasizes the results of the previously mentioned thematic analyses. However, it is recommended to conclude the discussion by mentioning possible limitations of the new proposed broken link model, which could be potential targets for futuristic applicable epidemic surges studies
The authors concluded their conclusion by focusing on the K-Value difference between Japan and other countries. In this part, there is a missed statement, which should be addressed through the authors regarding potential practical applications of the broken link model in futuristic studies and analyses.
I suggest however minor edits and revisions before publication in MDPI IJERPH:
Abstract:
p. 1 , row 14: I recommend using an alternative word to (consider), e.g. (think) or (presume)
Introduction:
p. 1 , row 23: I recommend using a less confirmative word than (accurately) e.g. (as accurate as possible)
p. 1 , row 31: I recommend using the word (corresponded) instead of (agreed)
p. 1 , row 41: Please use the word (whereas) instead of (where)
p.2 , row 46: Please use an alternative word to (microscopically) e.g. (comprehensively)
p. 2 , row 48: Please use the following (applied) instead of (apply)
Material and Methods:
p. 2 , row 66: Please use the following phrasing (as reported in Nishiura et al in their study of COVID-19 secondary transmission) instead of (as reported in Ref 7)
p. 2 , row 69: I recommend adding a concise description of secondary and higher transmission of COVID-19 before introducing the compartment models.
p. 2 , row 70: Please use the following (According to Nishiura et al) instead of (According to Ref 7)
p. 3 , row 75: I recommend using a different phrase to describe Figure 1 e.g. (A suggested formula of the broken link model) instead of (The idea to formulate the broken link model)
p. 3 , row 79-80: The following statement is very interesting and important to reemphasize (Contrary to the SIR model, the coupling between susceptible and infected
becomes time dependent in the broken-link model). I recommend adding the following statement into the discussion as well.
p. 3 , row 87-88: I recommend adding this as one of the limitation of the analysis despite the smaller contribution to the whole number of infected people
p. 3 , row 92, 94: As mentioned above in the general remarks, it would be recommended to add a list/table of definitions for the used indicators to help the flow of reading (e.g. N00: cumulative number of infected people in each infection wave generated by R0 , R0: the basic reproduction number, etc….)
p. 3 , row 99, 106: This final part represent a very important statement, which I recommend adding consicely into the discussion section.
Result:
p. 3 , row 111 : Please use the word (peak) instead of (bump)
p. 4 , row 120: The same here please use the word (three peaks) instead of (three bumps)
p. 6 , row 171: I recommend using the following phrase (corresponded to) instead of (is controlled by)
p. 6 , row 173: I recommend adding a concise definition of constant a
p. 7 , row 182: Please use the following phrasing (From Nakano et al) instead of (From Ref 3, )
p. 7 , row 190-191: I recommend removing the following statement, as it has no relevance to the broken link model analysis, and it creates confusion with the aforementioned statement (Thanks to the double vaccination, the 2nd and 3rd partial wave were suppressed to about 40% and 30%, respectively)
p. 7 , row 194: Please use the following word (showed) instead of (testified)
Conclusion:
p. 7 , row 222: I recommend adding the following phrase ( in the traditional used compartment model for COVID-19 spread analysis) after the word (COVID-19)
In addition to the above-detailed review. I would like to shed light on two points before proceeding to publication:
-The addition of a list or table of definitions of the used terms/indicators throughout the manuscript to help keep the flow of reading
-The Introduction section should be further supported to include essential information relating to background of the traditional COVID-19 compartment models of spread analysis (SIR/SEIR).
Author Response

(The authors gave the same response as above.)
